# Set Features for Anomaly Detection

**Niv Cohen**                                                            *nivc@cs.huji.ac.il*
*School of Computer Science and Engineering*
*The Hebrew University of Jerusalem, Israel*

**Issar Tzachor**                                              *issar.tzachor@mail.huji.ac.il*
*School of Computer Science and Engineering*
*The Hebrew University of Jerusalem, Israel*

**Yedid Hoshen**                                              *yedid.hoshen@mail.huji.ac.il*
*School of Computer Science and Engineering*
*The Hebrew University of Jerusalem, Israel*

**Reviewed on OpenReview:** `https://openreview.net/forum?id=aukOnn7J4M`

## Abstract

This paper proposes to use set features for detecting anomalies in samples that consist of unusual combinations of normal elements. Many leading methods discover anomalies by detecting an unusual part of a sample. For example, state-of-the-art segmentation-based approaches, first classify each element of the sample (e.g., image patch) as normal or anomalous and then classify the entire sample as anomalous if it contains anomalous elements. However, such approaches do not extend well to scenarios where the anomalies are expressed by an unusual combination of normal elements. In this paper, we overcome this limitation by proposing set features that model each sample by the distribution of its elements. We compute the anomaly score of each sample using a simple density estimation method, using fixed features. Our approach outperforms the previous state-of-the-art in image-level logical anomaly detection and sequence-level time series anomaly detection[1].

## 1 Introduction

Anomaly detection aims to automatically identify samples that exhibit unexpected behavior. In some detection tasks anomalies are quite subtle. For example, let us consider an image of a bag containing screws, nuts, and washers (Fig.1). There are two ways in which a sample can be anomalous: (i) one or more of the elements in the sample are anomalous. E.g., a broken screw. (ii) the elements are normal but appear in an anomalous combination. E.g., one of the washers might be replaced with a nut.

In recent years, remarkable progress has been made in detecting samples featuring anomalous elements. Segmentation-based methods were able to achieve very strong results on industrial inspection datasets (Bergmann et al., 2019). Such methods operate in two stages: First, we perform anomaly segmentation by detecting which (if any) of the elements of the sample are anomalous, e.g., by density estimation (Cohen & Hoshen, 2020; Defard et al., 2021; Roth et al., 2022). Given an anomaly segmentation map, we compute the sample-wise anomaly score as the number of anomalous elements, or the abnormality level of the most anomalous element. If the anomaly score exceeds a threshold, the entire sample is denoted as an anomaly. We denote this paradigm *detection-by-segmentation*.

Here, we tackle the more challenging case of detecting anomalies consisting of an unusual combination of normal elements. For example, consider the case where normal images contain two washers and two nuts, but

---

[1]Our code can be found at `https://github.com/NivC/SINBAD/`

anomalous images may contain one washer and three nuts. As each of the elements (nuts or washers) occur in natural images, detection-by-segmentation methods will not work. Instead, a more holistic understanding of the image is required to apply density estimation techniques. While simple global representations, such as taking the average of the representations of all elements might work in some cases, the result is typically too coarse to detect challenging anomalies.

Existing approaches tackle logical anomalies in images by reconstruction-based approaches - e.g., an autoencoder combining local and global classes (Bergmann et al., 2022; Batzner et al., 2023). These approaches have obtained strong results on some object types, while anomaly detection on other anomaly types remains low. Other approaches provide strong results on all object types but rely on additional supervision (Kim et al., 2024). For the analogues time-series task, where there is a global anomaly in the time series, the best-performing approach is currently a generalization-based approach (Qiu et al., 2021). We aim here to provide strong results on both tasks using a unified density estimation approach.

We propose to detect anomalies consisting of unusual combinations of normal elements using set representations. Our key insight, that *we should treat a sample as the set of its elements*, is driven by the assumption that in many cases the abnormality of a sample is more correlated with the distribution of elements than with their ordering. Each sample is therefore modeled as an orderless set of elements. The elements are represented using standard fixed feature embeddings, e.g., a deep representation extracted by a pre-trained neural network or handcrafted features. To describe this set of features we model their distribution as a set using a collection of histograms. We compute the histograms for a collection of random projection directions in feature space. The bin occupancies from all the histograms are concatenated together, forming our set representation. Finally, we score anomalies using density estimation on this set representation. We compare our set descriptor to previous approaches and highlight its connection to the sliced Wasserstein distance (SWD).

Our method, *SINBAD* (*Set IN*spection *B*ased *A*omalies *D*etection) is evaluated on two diverse tasks. The first task is image-level logical anomaly detection on the MVTec-LOCO datasets. We also evaluate our method on series-level time series anomaly detection. Our method outperforms more complex state-of-the-art methods while not using augmentations or training. Note that our method relies on the prior assumption that the elements are normal but their combination is anomalous. In scenarios where the elements themselves are anomalous, it is typically better to perform anomaly detection directly at the element level; E.g., detection-by-segmentation or other methods.

We make the following contribution:

- Identifying set representation as key for detecting anomalies consisting of normal elements.

- A novel set-based method for measuring the distance between samples.

- State-of-the-art results on logical and time series anomaly detection datasets.

## 2 Previous work

**Image Anomaly Detection.** A comprehensive review of anomaly detection can be found in Ruff et al. (2021). Early approaches such as Glodek et al. (2013); Latecki et al. (2007); Eskin et al. (2002) used handcrafted representations. Deep learning has provided a significant improvement on such benchmarks (Larsson et al., 2016; Ruff et al., 2018; Golan & El-Yaniv, 2018; Hendrycks et al., 2019; Ruff et al., 2019; Perera & Patel, 2019; Salehi et al., 2021; Tack et al., 2020). As density estimation methods utilizing pre-trained deep representation have made significant steps towards the supervised performance on such benchmarks (Deecke et al., 2021; Cohen & Avidan, 2022; Reiss et al., 2021; Reiss & Hoshen, 2021; Reiss et al., 2022); much research is now directed at other challenges (Reiss et al., 2022). Such challenges include detecting anomalous image parts which are small and fine-grained (Cohen & Hoshen, 2020; Li et al., 2021; Defard et al., 2021; Roth et al., 2022; Horwitz & Hoshen, 2022). The progress in anomaly detection and segmentation has been enabled by the introduction of appropriate datasets (Bergmann et al., 2019; 2021; Carrera et al., 2016; Jezek et al., 2021; Bonfiglioli et al., 2022). Recently, the MVTec-LOCO dataset Bergmann et al. (2022) has put the spotlight on fine-grained anomalies that cannot be identified using single patches, but can only be

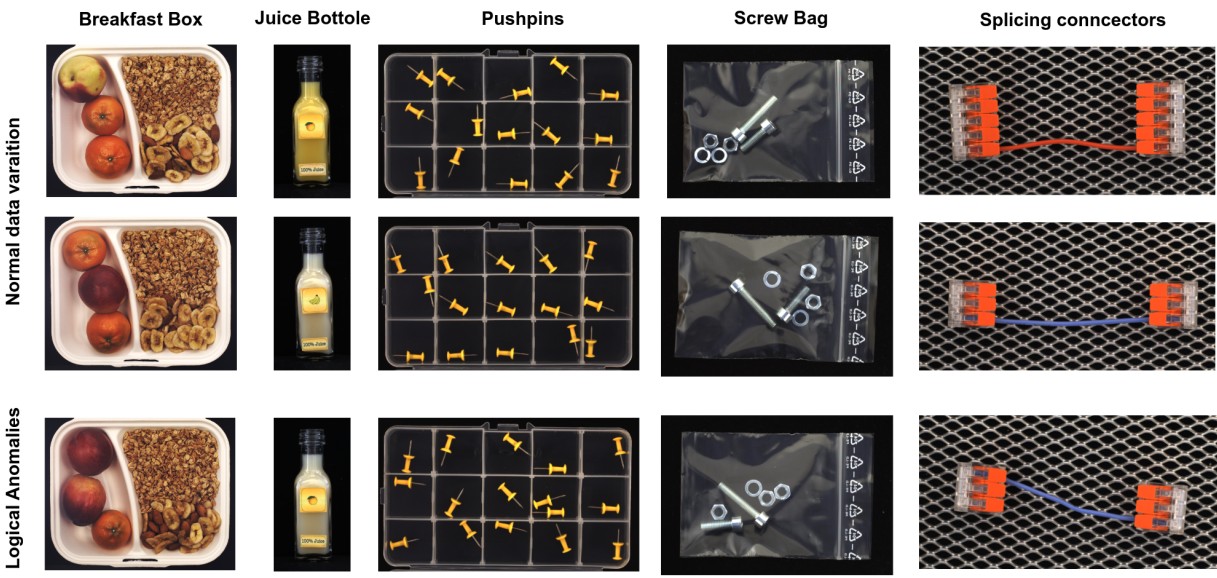

Figure 1: In set anomalies, each image element (e.g., patch) may be normal even when their combination is anomalous. This is challenging as the variation in the normal data may be higher than between normal and anomalous elements (e.g., swapping a bolt and a washer in the *screw bag* class).

identified when examining the connection between different (otherwise normal) elements in an image. Here, we will focus on detecting such *logical* anomalies.

**Time series Anomaly detection.** A general review on anomaly detection in time series can be found in (Blázquez-García et al., 2021). In this paper, we are concerned with anomaly detection of entire sequences, i.e., cases where an entire signal may be abnormal. Traditional approaches for this task include generic anomaly detection approaches such as $k$ nearest neighbors ($k$NN) based methods, e.g., vanilla $k$NN Eskin et al. (2002) and Local Outlier Factor (LOF) Breunig et al. (2000), tree-based methods Liu et al. (2008), one-class classification methods Tax & Duin (2004) and SVDD Schölkopf et al., and auto-regressive methods that are particular to time series anomaly detection Rousseeuw & Leroy (2005). With the advent of deep learning, the traditional approaches were augmented with deep-learned features: Deep one-class classification methods include DeepSVDD Ruff et al. (2018) and DROCC Goyal et al. (2020). Deep auto-regressive methods include RNN-based prediction and auto-encoding methods Bontemps et al. (2016); Malhotra et al. (2016). In addition, some deep learning anomaly detection approaches are conceptually different from traditional approaches. These methods use classifiers trained on normal data, assuming they will struggle to generalize to anomalous data (Bergman & Hoshen, 2020; Qiu et al., 2021).

**Discretized Projections.** Discretized projections of multivariate data have been used in many previous works. Locally sensitive hashing Dasgupta et al. (2011) uses random projection and subsequent binary quantization as a hash for high-dimensional data. It was used to facilitate fast $k$ nearest neighbor search. Random projections transformation is also highly related to the Radon transform Radon (1917). Kolouri et al. Kolouri et al. (2015) used this representation as a building block in their set representation. HBOS Goldstein & Dengel (2012) performs anomaly detection by representing each dimension of multivariate data using a histogram of discretized variables. Rocket and mini-rocket Dempster et al. (2020; 2021) represent time series for classification using the averages of their window projection. LODA Pevnỳ (2016) extends this work, by first projecting the data using a random projection matrix. We differ from LODA in the use of a different density estimator and in using sets of multiple elements rather than single sample descriptions.

## 3 Set Features for Anomaly Detection

### 3.1 A Set is More Than the Sum of its Parts

Detecting anomalies in complex samples consisting of collections of elements requires understanding how the different elements of each sample interact with one another. As a motivating example let us consider the *screw bag* class from the MVTec-LOCO dataset (Fig. 1). Each normal sample in this class contains two screws (of different lengths), two nuts, and two washers. Anomalies may occur, for example, when an additional nut replaces one of the washers. Detecting anomalies such as these requires describing all elements within a sample together, since each local element on its own could have come from a normal sample.

A typical way to aggregate element descriptor features is by average pooling - taking the average of the features describing each element. Yet, this is not always suitable for set anomaly detection. In supervised learning, average pooling is often built into architectures such as ResNet He et al. (2016) or DeepSets Zaheer et al. (2017), in order to aggregate local features. Therefore, deep features learned with a supervised loss are already trained to be effective for pooling. However, for lower-level feature descriptors this may not be the case. As demonstrated in Fig.2, the average of a set of features is far from a complete description of the set. This is especially true in anomaly detection, where density estimation approaches require more discriminative features than those needed for supervised learning (Reiss et al., 2022). Even when an average pooled set of features works for a supervised task, it might not work for anomaly detection.

Therefore, we choose to model a set by the distribution of its elements in the embedded feature space, ignoring the ordering between them. A naive way of doing so is using a discretized, volumetric representation, similar to 3D voxels for point clouds. Unfortunately, such approaches cannot scale to high dimensions, and more compact representations are required. We choose to represent sets using a collection of 1D histograms. Each histogram represents the density of the elements of the set when projected along a particular direction. We take the bin occupancies of such histograms as our features. We provide an illustration of this idea in Figure 2.

In some cases, projecting a set along its original axes may not be discriminative enough. Histograms along the original axes correspond to 1D marginals, and may map distant elements to the same histogram bins (see Fig.2 for an illustration). On the other side, we can see at the bottom of the figure that when the set elements are first projected along another direction, the histograms of the two sets are distinct. This suggests a set description method: first project each set along a shared random direction and then compute a 1D histogram for each set along this direction. We can obtain a more powerful descriptor by repeating this procedure with projections along multiple random directions. We benchmark this approach in section 3.5.

### 3.2 Preliminaries

We are provided a training set $\mathcal{S}$ containing a set of $N_S$ samples, we denote a sample as $x \in \mathcal{S}$. We assume that all the training samples are normal. We wish to learn a model that operates on a new, test sample $\tilde{x}$ and outputs an anomaly score. We label samples with anomaly scores higher than a predetermined threshold value as anomalies. The unique aspect of our method is that it treats each sample $x$ as consisting of a set of $N_E$ elements, where we denote each element as $e \in x$. Examples of such elements include patches for images, or temporal windows for time series. We assume the existence of a powerful feature extractor $F$ that maps each raw element $e$ into an element feature descriptor $F(e)$. We will describe specific implementations of the feature extraction for two important applications: images and time series, in section 4.

### 3.3 Set Features by Histogram of Projections

Motivated by the toy example in section 3.1, we propose to model each set by the histogram of the values of its elements along a collection of directions. We provide an algorithm box Alg.1 summarizing our steps.

**Feature extractor.** We split each sample to elements $e \in x$ and extract a feature representation for each $\{F(e) \mid e \in x\}$. We describe the implementation of $F$ in Sec.4 as it differs for the time series and image modalities.

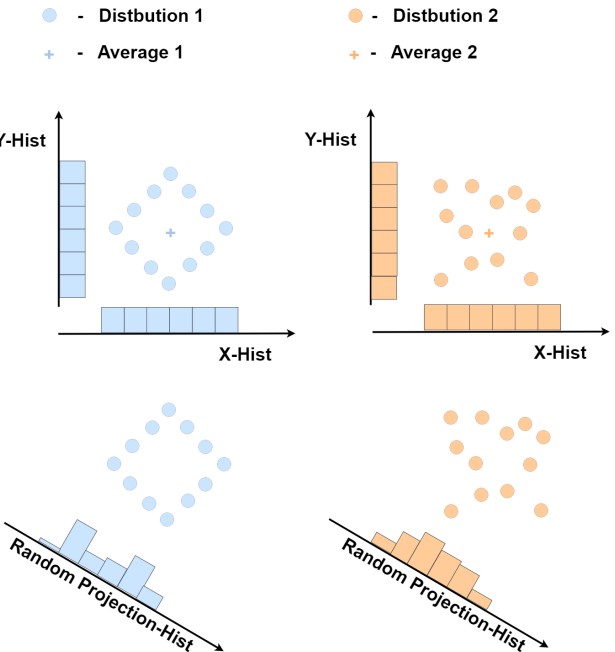

Figure 2: Random projection histograms allow us to distinguish between sets where other methods could not. The two sets are similar in their averages and histograms along the original axes, but result in different histograms when projected along a random axis.

**Histogram descriptor.** While average pooling the features of all elements in the set is an obvious set descriptor, it may result in insufficiently informative representations (section 3.1). Instead, we describe the set by computing the set's histogram for each feature dimension and concatenating them.

For each sample, we denote the set of values for the $j^{\text{th}}$ dimension of the feature embeddings as $s[j] = \{F(e)[j] \mid e \in x\}$. Note that each set $s[j]$ consists of $N_E$ scalar elements coming for each sample. We compute the maximal and minimal values of $s[j]$ for each dimension $j$, among all the elements from all samples combined $(N_S \cdot N_E)$, and divide the region between them into $K$ bins. Finally, we compute histograms $h[j]$ for each of the $N_D$ dimensions, describing the set $s[j]$, and concatenate each histogram of $K$ bins of each histogram to a single set descriptor $h \in \mathbb{R}^{N_D \cdot K}$.

**Projection.** As discussed before, not all projection directions are equally informative for describing the distributions of sets. In the general case, it is unknown which directions are the most informative for capturing the difference between normal and anomalous sets. As we cannot tell the best projection directions in advance, we randomly project the features. This minimizes the likelihood of all projections being in catastrophically poor directions, such as those illustrated in Fig. 2.

In practice, we generate a random projection matrix $P \in \mathbb{R}^{N_D \times N_P}$ by sampling values for each dimension from the Gaussian distribution $\mathbb{N}(0, 1)$. We project the features of each element of $x$, yielding projected features $f$:

$$f = P \cdot F(e) \tag{1}$$

We run the histogram descriptor procedure described above on the projected features. The final set descriptor $h \in \mathbb{R}^{N_P \cdot K}$ is the concatenation of the $N_P$ histograms.

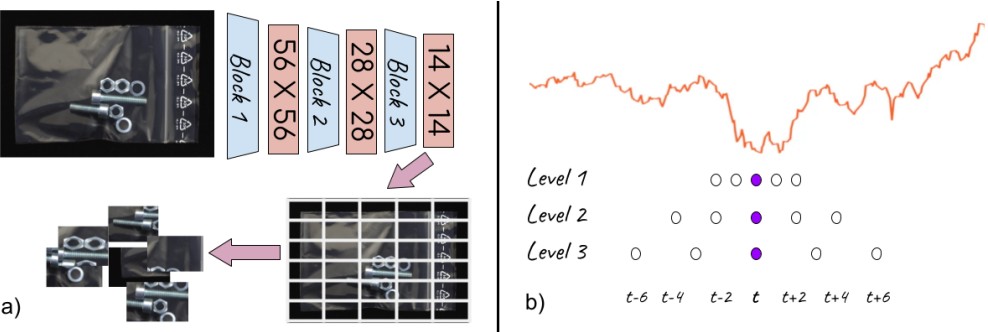

Figure 3: For both image and time series samples, we extract set elements at different granularity. For images (left), the sets of elements are extracted from different ResNet levels. For time series (right), we take pyramids of windows at different strides around each time step.

### 3.4    Anomaly Scoring

We perform density estimation on the set descriptors, expecting unusual test samples to have unusual descriptors, far from those of the normal train set. We define the anomaly score as the Mahalanobis distance, the negative log-likelihood in feature space. We denote the mean and covariance of the histogram projection features of the normal data as $\mu$ and $\Sigma$:

$$a(h) = (h - \mu)^T \Sigma^{-1} (h - \mu) \tag{2}$$

### 3.5    Connection to Previous Set Descriptors and the Wasserstein Distance

**Classical set descriptors.** Many prior methods have been used to describe sets of image elements, among them Bag-of-Features Csurka et al. (2004), VLAD Jégou et al. (2010), and Fisher-Vectors Sánchez et al. (2013). These methods begin with a preliminary clustering stage (K-means or Gaussian Mixture Model). They then describe the set using the zeroth, first, or second moments of each cluster. The comparison in Appendix B shows that our method outperforms clustering-based methods in describing our feature sets.

**Wasserstein distance.** Our method is closely related to the Wasserstein distance, which measures the minimal distance required to transport the probability mass from one distribution to the other. As computing the Wasserstein distance for high-dimensional data such as ours is computationally demanding, the Sliced Wasserstein Distance (SWD) Bonneel et al. (2015), was proposed as an alternative. The $SWD_1$ between two sets, $x$ and $y$, has a particularly simple form:

$$SWD_1(x, y) = \|h_{Px} - h_{Py}\|_1 \tag{3}$$

where $h_{Px}, h_{Py}$ are the random projections histogram of sets $x$ and $y$, that we defined in Sec.3.3.

As the histogram projections have a high correlation between them, it is necessary to decorrelate them. This is done here using a Gaussian model. We refer to Appendix G for an explanation of why a Gaussian model is appropriate here. The Mahalanobis distance used here therefore performs better than the simple $SWD_1$ distance. While this weakens the connection to the Wasserstein distance, this was crucial for most time-series datasets (see Table 12). In practice, we opted to use $k$NN with the Mahalanobis distance rather than simply computing the Mahalanobis distance to $\mu$ as it worked slightly better (see Appendix B).

We note that the Sliced Wasserstein distance can be calculated directly, without using the suggested histogram binning. However, our histogram-based approach is necessary for our density estimation method. Namely, to compute $k$NN using the Mahalanobis distance, we require a feature representation for each point, rather than

relying just on a pairwise distance function between samples. We compare our approach directly with the precise and binned Sliced Wasserstein distance in Appendix F.

---

**Algorithm 1** Set-based Anomaly Detection with Histogram Projections

1: **Input:** Training set $\mathcal{S}$ with $N_S$ normal samples, feature extractor $F$, feature dimension $N_D$, number of projections $N_P$, number of histogram bins $K$
2: **Output:** An anomaly detection score
3: Generate random projection matrix $P \in \mathbb{R}^{N_D \times N_P}$
4: **for** each sample $x \in \mathcal{S}$ **do**
5:     **for** each element $e \in x$ **do**
6:         Extract features $F(e)$
7:         Project features: $f = P \cdot F(e)$
8:     **end for**
9:     **for** each projection dimension $j = 1$ to $N_P$ **do**
10:         Build and histogram for each projected dimension, each with $K$ bins
11:         For each sample $x$ we populate the histogram $h_x[j]$ of projected features in the $j'th$ dimension with the projected features $f[j]$     ▷ Each histogram is populated with $N_E$ values, as the number of elements extracted from each sample
12:     **end for**
13:     Concatenate histograms for each sample: $h_x = [h_x[1], \ldots, h_x[N_P]]$   ▷ resulting dimension is $K \cdot N_P$
14: **end for**
15: Compute mean $\mu$ and covariance $\Sigma$ of $h_S$ (the set $h_S = \{h_x \mid x \in S\}$)
16: **for** test sample $\tilde{x}$ **do**
17:     Compute $h_{\tilde{x}}$ as above
18:     Calculate anomaly score: $a(h_{\tilde{x}}) = (h_{\tilde{x}} - \mu)^T \Sigma^{-1} (h_{\tilde{x}} - \mu)$
19: **end for**

---

## 4 Application to Image and Time Series Anomaly Detection

### 4.1 Images as Sets

Images can be seen as consisting of a set of elements at different levels of granularity. This ranges from pixels to small patches, to low-level elements such as lines or corners, up to high-level elements such as objects. For anomaly detection, we typically do not know in advance the correct level of granularity for separating between normal and anomalous samples (Heckler et al., 2023). The correct level may depend on the anomalies we will encounter, which are unknown during training. Instead, we use multiple levels of granularity, describing image patches of different sizes, and combine their scores.

In practice, we use representations from intermediate blocks of a pre-trained ResNet (He et al., 2016). As a ResNet network simultaneously embeds many local patches of each image, we pass the image samples through the network encoder and extract our representations from the intermediate activations at the end of different ResNet blocks (see Fig.3). We define each spatial location in the activation map as an element. Note that as different blocks have different resolutions, they yield different numbers of elements per layer. We run our set methods with the elements at the end of each residual block used and combine the results in an ensemble as detailed in the appendix (App.C.1).

### 4.2 Time Series as Sets

Time series data can be viewed as a set of temporal windows. Similarly to images, it is generally not known in advance which temporal scale is relevant for detecting anomalies. I.e., what is the duration of windows which includes the semantic phenomenon. Inspired by *Rocket* Dempster et al. (2020), we define the basic elements of a time series as a collection of temporal window pyramids. Each pyramid contains $L$ windows. All the windows in a pyramid are centered at the same time step, each containing $\tau$ samples (Fig.3). The first level window includes $\tau$ elements with stride 1, the second level window includes $\tau$ elements with stride

Table 1: Anomaly detection on MVTec-LOCO. ROC-AUC (%). See Tab.6 for the full table.

|  |  | f-AnoGAN | MNAD | ST | SPADE | PCore | GCAD | SINBAD |
|---|---|---|---|---|---|---|---|---|
| Logical Ano. | Breakfast box | 69.4 | 59.9 | 68.9 | 81.8 | 77.7 | 87.0 | **97.7 ± 0.2** |
|  | Juice bottle | 82.4 | 70.5 | 82.9 | 91.9 | 83.7 | **100.0** | 97.1 ± 0.1 |
|  | Pushpins | 59.1 | 51.7 | 59.5 | 60.5 | 62.2 | **97.5** | 88.9 ± 4.1 |
|  | Screw bag | 49.7 | 60.8 | 55.5 | 46.8 | 55.3 | 56.0 | **81.1 ± 0.7** |
|  | Splicing connectors | 68.8 | 57.6 | 65.4 | 73.8 | 63.3 | 89.7 | **91.5 ± 0.1** |
|  | Avg. Logical | 65.9 | 60.1 | 66.4 | 71.0 | 69.0 | 86.0 | **91.2 ± 0.8** |
| Structural Ano. | Breakfast box | 50.7 | 60.2 | 68.4 | 74.7 | 74.8 | 80.9 | **85.9 ± 0.7** |
|  | Juice bottle | 77.8 | 84.1 | **99.3** | 84.9 | 86.7 | 98.9 | 91.7 ± 0.5 |
|  | Pushpins | 74.9 | 76.7 | **90.3** | 58.1 | 77.6 | 74.9 | 78.9 ± 3.7 |
|  | Screw bag | 46.1 | 56.8 | 87.0 | 59.8 | 86.6 | 70.5 | **92.4 ± 1.1** |
|  | Splicing connectors | 63.8 | 73.2 | **96.8** | 57.1 | 68.7 | 78.3 | 78.3 ± 0.3 |
|  | Avg. Structural | 62.7 | 70.2 | **88.3** | 66.9 | 78.9 | 80.7 | 85.5 ± 0.7 |
|  | Avg. Total | 64.3 | 65.1 | 77.4 | 68.9 | 74.0 | 83.4 | **88.3 ± 0.7** |

Table 2: Anomaly detection on MVTec-LOCO. ROC-AUC (%). See Tab.6 for the full table.

|  | SINBAD | EfficientAD (reported) | EfficientAD (reproduced) | PUAD | SINBAD+EfficientAD |
|---|---|---|---|---|---|
| Logical | 91.2 ± 0.8 | 86.8 | 85.9 ± 0.4 | 92.0 | **92.7 ± 0.6** |
| Structural | 85.5 ± 0.7 | 94.7 | 93.8 ± 0.4 | 94.1 | **95.8 ± 0.5** |
| All | 88.3 ± 0.7 | 90.8 | 89.8 ± 0.5 | 93.1 | **94.2 ± 0.6** |

2, etc. Such a window pyramid is computed for each time step in the series. The entire series is represented as a set of pyramids of its elements. Implementation details for both modalities are described in Sec.C.2.

## 5 Results

### 5.1 Logical Anomaly Detection Results

**Logical Anomalies Dataset.** We use the recently published MVTec-LOCO dataset Bergmann et al. (2022) to evaluate our method's ability to detect anomalies caused by unusual configurations of normal elements. This dataset features five different classes: *breakfast box*, *juice bottle*, *pushpins*, *screw bag* and *splicing connector* (see Fig.1). Each class includes: (i) a training set of normal samples ($\sim 350$ samples). (ii) a validation set, containing a smaller set of normal samples ($\sim 60$ samples). (iii) a test set, containing normal samples, structural anomalies, and logical anomalies ($\sim 100$ each).

The anomalies in each class are divided into *structural anomalies* and *logical anomalies*. Structural anomalies feature local defects, somewhat similar to previous datasets such as Bergmann et al. (2019). Conversely, logical anomalies may violate 'logical' conditions expected from the normal data. As one example, an anomaly may include a different number of objects than the numbers expected from a normal sample (while all the featured object types exist in the normal class; see Fig.1). Other types of logical anomalies in the dataset may include cases where distant parts of an image must correlate with one another. For instance, within the normal data, the color of one object may correlate with the length of another object. These correlations may break in an anomalous sample.

**Baselines.** We compare to baseline methods used by the paper which presented the MVTec-LOCO dataset Bergmann et al. (2022): *Variational Model (VM)* Steger (2001), *MNAD*, *f-AnoGAN* Schlegl et al. (2017), *AE / VAE. Student Teacher* (ST), *SPADE*, *PatchCore* (PCore) Roth et al. (2022). We also compare to *GCAD* Bergmann et al. (2022) - a reconstruction-based method, based on both local and global deep ResNet features, which was explicitly designed for logical anomaly detection; *EfficientAD* - Batzner et al. (2023), a reconstruction-based method with a loss aimed at preventing an autoencoder from reconstructing well anomalous unseen images. We also report the results by *PUAD* Sugawara & Imamura (2024), an ensemble

Table 3: Anomaly detection on the UEA datasets, average ROC-AUC (%) over all classes. See Tab.7 for the complete table with error bounds.

|      | OCSVM | IF   | RNN  | ED   | DSVDD | DAG  | GOAD | DROCC | NeuTraL | Ours     |
|------|-------|------|------|------|-------|------|------|-------|---------|----------|
| EPSY | 61.1  | 67.7 | 80.4 | 82.6 | 57.6  | 72.2 | 76.7 | 85.8  | 92.6    | **98.1** |
| NAT  | 86.0  | 85.4 | 89.5 | 91.5 | 88.6  | 78.9 | 87.1 | 87.2  | 94.5    | **96.1** |
| SAD  | 95.3  | 88.2 | 81.5 | 93.1 | 86.0  | 80.9 | 94.7 | 85.8  | **98.9**| 97.8     |
| CT   | 97.4  | 94.3 | 96.3 | 79.0 | 95.7  | 89.8 | 97.7 | 95.3  | 99.3    | **99.7** |
| RS   | 70.0  | 69.3 | 84.7 | 65.4 | 77.4  | 51.0 | 79.9 | 80.0  | 86.5    | **92.3** |
| Avg. | 82.0  | 81.0 | 86.5 | 82.3 | 81.1  | 74.6 | 87.2 | 86.8  | 94.4    | **96.8** |

method combining Batzner et al. (2023) and Rippel et al. (2021). Finally, we report *SINBAD+EfficientAD*, a simple average of EfficientAD's per-sample results and ours. As the last set of baselines does not always report per-class accuracies, we report them in a different table.

**Metric.** Following the standard metric in image-level anomaly detection we use the ROC-AUC metric.

**Results.** We report per-class results on image-level detection of logical anomalies and structural anomalies in Tab.1. Interestingly, we find complementary strengths between our approach and GCAD, a reconstruction-based approach by Bergmann et al. (2022). Although GCAD performed better on specific classes (e.g., *pushpins*), our approach provides better results on average. Notably, our approach provides non-trivial anomaly detection capabilities on the *screw bag* class, while baseline approaches are close to the random baseline performance (which is 50% ROC-AUC). EfficientAD Batzner et al. (2023), focuses on structural anomalies and achieves impressive results on them, but underperforms on logical anomalies (see Tab.2).

As our method is comparatively strong on specific classes (e.g., *Screw bag, Logical*, 81.1% compared to 56.0% of GCAD and 55.5% of EfficientAD), it serves as a valuable component in ensemble methods. For example, a simple combination of our method with EfficientAD Batzner et al. (2023) (*SINBAD+EfficientAD*) outperforms all other methods and ensembles (See table 2).

Our approach also improves upon detection-by-segmentation methods in detecting structural anomalies in some classes. This is somewhat surprising, as one may assume that detection-by-segmentation approaches would perform well in these cases. One possible reason for that is the high variability of the normal data in some of the classes (e.g., *breakfast box*, *screw bag*, Fig.1). This high variability may induce false positive detections for baseline approaches. While different methods provide complementary strengths, on average, our method provides state-of-the-art results in logical anomaly detection. See also the discussion at Sec.6.

## 5.2 Time Series anomaly Detection Results

**Time series dataset.** We evaluate on the five datasets used in NeurTraL-AD Qiu et al. (2021): *RacketSports (RS)*. Accelerometer and gyroscope recordings of players playing different racket sports. Each sport is designated as a class. *Epilepsy (EPSY)*. Accelerometer recording of healthy actors simulating four activity classes, e.g. an epileptic shock. *Naval air training and operating procedures standardization (NAT)*. Positions of sensors mounted on body parts of a person performing activities. There are six different activity classes in the dataset. *Character trajectories (CT)*. Velocity trajectories of a pen on a tablet. There are 20 characters in this dataset. *Spoken Arabic Digits (SAD)*. MFCC features ten Arabic digits spoken by 88 speakers.

**Baselines.** We compare the results of several baseline methods reported by Qiu et al. (2021). The methods cover the following paradigms: *One-class classification*: One-class SVM (OC-SVM), and its deep versions, DeepSVDD ("DSVDD") Ruff et al. (2018), DROCC Goyal et al. (2020). *Tree-based detectors*: Isolation Forest (IF) Liu et al. (2008). *Density estimation*: LOF, a version of nearest neighbor anomaly detection Breunig et al. (2000). DAGMM ("DAG") Zong et al. (2018): density estimation in an auto-encoder latent space. *Auto-regressive methods* - RNN and LSTM-ED ("ED") - deep neural network-based version of auto-regressive prediction models Malhotra et al. (2016). *Transformation prediction* - GOAD Bergman & Hoshen (2020) and

NeuTraL-AD Qiu et al. (2021) are based on transformation prediction, and are adaptations of RotNet-based approaches such as GEOM Golan & El-Yaniv (2018).

**Metric.** Following Qiu et al. (2021), we use the series-level ROC-AUC metric.

**Results.** Our results are presented in Tab. 3. We can observe that different baseline approaches are effective for different datasets. $k$NN-based LOF is highly effective for SAD which is a large dataset but achieves worse results for EPSY. Auto-regressive approaches achieve strong results on CT. Transformation-prediction approaches, GOAD and NeuTraL achieve the best performance of all the baselines. The learned transformations of NeuTraL achieved better results than the random transformations of GOAD. Our method achieves the best overall results both on average and individually on all datasets apart from SAD, where it is comparable but a little lower than NeuTraL. We note that unlike NeuTraL, our method is far simpler, does not use deep neural networks, and is very fast to train and evaluate. It also has fewer hyperparameters.

### 5.3 Implementation Details

We provide here the main implementation details for our image anomaly detection application. Further implementation details for the image application can be found in App.C.1; Implementation details for the time series application can be found in the supplementary material in App.C.2.

**ResNet levels.** We use the representations from the 3rd and 4th blocks of a *WideResNet50×2* (resulting in sets size $7 \times 7$ and $14 \times 14$ elements, respectively). We also use all the raw pixels in the image as an additional set (resized to $224 \times 224$ elements). The total anomaly score is the average of the anomaly scores obtained for the set of 3rd ResNet block features, the set of 4th ResNet block features, and the set of raw pixels. The average anomaly score is weighted by the following factors $(1, 1, 0.1)$ respectively (see App.D for our robustness to the choice of weighting factor).

**Multiple crops for image anomaly detection.** Describing the entire image as a single set might sometimes lose discriminative power when the anomalies are localized. To mitigate this issue, we can treat only a part of an image as our entire set. To do so, we crop the image to smaller images by a factor of $c$, and compare the elements taken from each crop. We compute an anomaly score for each crop scale and for each center location. We then average over the anomaly scores of the different crop center locations for the same crop scale $c$. Finally, for each ResNet level, we average the anomaly scores over the different crop scales. We use crop scales of $\{1.0, 0.7, 0.5, 0.33\}$. The different center locations are taken with a stride of 0.25 of the entire image. We observe that combining different scales offers only a marginal advantage (see Tab.4), while baseline methods require significantly more forward passes through the feature extractors Roth et al. (2022).

**Runtime.** A simple implementation of our method can run in real time ($> 20$ images per second) without multiple crops. Using multiple crops can be simply parallelized on multiple GPUs.

### 5.4 Ablations

We present ablations for the image logical AD methods. For further ablations of the histogram parameters and for the time series application, see appendix F.

**Using individual ResNet levels.** In Tab.4 we report the results when different components of our multi-level ResNet ensemble are removed. We report the results using only the representation from the third or fourth ResNet block ("Only 3" / "Only 4"). We also report the results using both ResNet blocks but without the raw-pixels level ("No Pixels").

**No multiple crops ablation.** We also report our results without the multiple crops ensemble (described in Sec.5.3). We feed only the entire image for the set extraction stage ("Only full"). As expected, using multiple receptive fields is beneficial for classes where small components are important to determine abnormality.

**Ablating our histogram density-estimation method.** In Tab.5 we ablate different aspects of our histogram set descriptors. *Sim. Avg.* We show a simple averaging Lee et al. (2018) of the set features (see also Fig. 2), ablating our entire set-features approach. This yields a significantly worse performance. *No proj.* We ablate our use of random projections (Sec.3.3). We replace the random histograms with similar

Table 4: Ablation for logical image AD. Logical Anomalies. ROC-AUC (%).

|  | Only 3 | Only 4 | No pixels | Only full | Ours |
|---|---|---|---|---|---|
| Breakfast box | 95.9 | 95.7 | 96.8 | 97.2 | **97.7** |
| Juice bottle | 93.0 | 97.0 | 95.8 | 97.0 | **97.1** |
| Pushpins | 79.2 | 67.0 | 74.0 | **89.9** | 88.9 |
| Screw bag | 79.8 | 70.4 | 76.6 | 76.2 | **81.1** |
| Splicing connectors | 84.7 | 85.6 | 86.1 | 90.7 | **91.5** |
| Average | 86.5 | 83.1 | 85.9 | 90.2 | **91.2** |

Table 5: MVTec-LOCO ablation (no raw-pixels). Logical Anomalies. ROC-AUC (%).

|  | Sim. Avg. | No Proj. | No Whit. | Ours |
|---|---|---|---|---|
| Breakfast box | 84.6 | 91.7 | 95.9 | **97.0** |
| Juice bottle | **98.0** | 97.3 | 97.5 | 96.2 |
| Pushpins | 63.5 | 69.3 | 73.4 | **73.7** |
| Screw bag | 65.0 | 68.2 | 72.5 | **77.5** |
| Splicing connectors | 87.4 | 84.5 | **87.9** | 85.9 |
| Average | 79.7 | 82.2 | 85.5 | **86.1** |

histograms using the raw given features. *No whit.* We ablate our Gaussian modeling of the set features. The whitening is not essential for the image modality, as it is for the time-series data (see Tab.12). We compare these variants of our method using the 3rd and 4th ResNet blocks, as the raw pixels level adds significant variance overshadowing the difference between some of the alternatives. While ablation may give stronger results in specific cases, our set approach together with the random projections and whitening generally outperforms.

**Ablating the number of bins and the number of projections.** While generally we would like to have as many random projections as possible; and a large number of bins per histogram, as long we have enough statistics to estimate the occupancy in each of them; We find that in practice the values we choose are large enough. We show in the App. Tab.10,11 that while significantly lower values in these parameters degrade our performance, the benefit from using larger values saturates.

## 6 Discussion

**Complementary strength of density estimation and reconstruction based approaches for logical anomaly detection.** Our method and GCAD Bergmann et al. (2022), a reconstruction-based approach, exhibit complementary strengths. Our method is most suited to detect anomalies resulting from the distribution of featured objects in each image. E.g., object replacements, additional or missing objects, or components indicating a logical inconsistency with the rest of the image. The generative modeling by GCAD gives stronger results when the positions of the objects are anomalous. E.g., one object containing another when it should not, or vice versa, as in the *Pushpins* class. The intuition here is that our approach treats the patches as an unordered set, and might not capture exact spatial relations between the objects. Therefore, it may be a natural direction to try and use both approaches together. A practical way to take advantage of both approaches would be an ensemble as we suggest in Sec.5. Ultimately, future research is likely to lead to the development of better approaches, combining the strengths of both methods.

**Relation to previous random projection methods.** Our method is related to several previous methods. HBOS Goldstein & Dengel (2012) and LODA Pevný (2016) also used similar projection features for anomaly detection. Yet, these methods perform histogram-based density, ignoring the dependency across projections. As they can only be applied to a single element, they do not achieve competitive performance for time series AD. Rocket and mini-rocket Dempster et al. (2020; 2021) also average projection features across selected windows from a given sample but do not apply to image data.

Additional discussion points can be found in App.H.

## 7 Limitations

**Detecting structural anomalies.** Our approach aims to detect specific, yet important, types of anomalies - image-level logical anomalies and the analogues time-series sequence-level anomalies. It is not particularly effective for detecting local structural anomalies, such as scratches or dents in images of objects Bergmann et al. (2019); Zou et al. (2022), object-level anomalies Reiss et al. (2021), or local time-series anomalies Blázquez-García et al. (2021). We evaluate our method on appropriate datasets. Currently, only one dataset evaluates logical anomalies Bergmann et al. (2022). Yet, this very comprehensive dataset contains 5 different sub-tasks, where each sub-task features numerous different types of anomalies. An anomaly detection method must rely on some assumptions regarding the nature of the anomalies one wishes to discover (Reiss et al., 2023). Therefore, when the type of anomalies is unknown, we recommend combining our method with methods tailored to different types of anomalies (Reiss et al., 2022; Roth et al., 2022; Batzner et al., 2023).

**Element-level anomaly detection.** Our method focuses on sequence-level time series and image-level anomaly detection. In some applications, a user may also want a segmentation map of the most anomalous elements of each sample. We note that for logical anomalies, this is often not well defined. E.g., when we have an image with 3 nuts as opposed to the normal 2, each of them may be considered anomalous. To provide element-level information, our method can be combined with current segmentation approaches by incorporating the knowledge of a global anomaly. E.g., removing false positive segmentations if an image is normal. Directly applying our set features for anomaly segmentation is left for future research.

**Class-specific performance.** In some classes we do not perform as well compared to baseline approaches. A better understanding of the cases where our method fails would be beneficial for deploying it in practice.

## 8 Conclusion

We presented a method for detecting anomalies caused by unusual combinations of normal elements. We introduce set features dedicated to capturing such phenomena and demonstrate their applicability for images and time series. Extensive experiments established the strong performance of our method. As with any anomaly detection method, our approach is biased to detect some abnormality modes rather than others. Using a few anomaly detection methods together may allow enjoying their complimentary benefits, and is advised in many practical cases.

## 9 Acknowledgment

This research was partially supported by the Israeli Science Foundation, the Israeli Council for Higher Education (VATAT), the Hebrew University Data Science grants (CIDR), the Malvina and Solomon Pollack scholarship, and a grant from KLA. Niv Cohen was partially supported by the Israeli data science scholarship for outstanding postdoctoral fellows (VATAT). We thank Paul Bergmann for kindly sharing numerical results for many of the methods compared on the MVTec-LOCO dataset.

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

## Contents

## A   Full Results Tables

The full table for the image logical anomaly detection experiment can be found in Tab.6. The full table for the time series anomaly detection experiments including error bounds for our method and baselines that reported them can be found in Tab.7. The difference between the methods is significantly larger than the standard error.

Table 6: Anomaly detection on the MVTec-LOCO dataset. ROC-AUC (%).

| | | VM | AE | VAE | f-AG | MNAD |
|---|---|---|---|---|---|---|
| **Logical Anom.** | Breakfast box | 70.3 | 58.0 | 47.3 | 69.4 | 59.9 |
| | Juice bottle | 59.7 | 67.9 | 61.3 | 82.4 | 70.5 |
| | Pushpins | 42.5 | 62.0 | 54.3 | 59.1 | 51.7 |
| | Screw bag | 45.3 | 46.8 | 47.0 | 49.7 | 60.8 |
| | Splicing connectors | 64.9 | 56.2 | 59.4 | 68.8 | 57.6 |
| | Avg. Logical | 56.5 | 58.2 | 53.8 | 65.9 | 60.1 |
| **Structural Ano.** | Breakfast box | 70.1 | 47.7 | 38.3 | 50.7 | 60.2 |
| | Juice bottle | 69.4 | 62.6 | 57.3 | 77.8 | 84.1 |
| | Pushpins | 65.8 | 66.4 | 75.1 | 74.9 | 76.7 |
| | Screw bag | 37.7 | 41.5 | 49.0 | 46.1 | 56.8 |
| | Splicing connectors | 51.6 | 64.8 | 54.6 | 63.8 | 73.2 |
| | Avg. Structural | 58.9 | 56.6 | 54.8 | 62.7 | 70.2 |
| | Avg. Total | 57.7 | 57.4 | 54.3 | 64.3 | 65.1 |
| | | ST | SPADE | PCore | GCAD | SINBAD |
| **Logical Anom.** | Breakfast box | 68.9 | 81.8 | 77.7 | 87.0 | **97.7 ± 0.2** |
| | Juice bottle | 82.9 | 91.9 | 83.7 | **100.0** | 97.1 ± 0.1 |
| | Pushpins | 59.5 | 60.5 | 62.2 | **97.5** | 88.9 ± 4.1 |
| | Screw bag | 55.5 | 46.8 | 55.3 | 56.0 | **81.1 ± 0.7** |
| | Splicing connectors | 65.4 | 73.8 | 63.3 | 89.7 | **91.5 ± 0.1** |
| | Avg. Logical | 66.4 | 71.0 | 69.0 | 86.0 | **91.2 ± 0.8** |
| **Structural Ano.** | Breakfast box | 68.4 | 74.7 | 74.8 | 80.9 | **85.9 ± 0.7** |
| | Juice bottle | **99.3** | 84.9 | 86.7 | 98.9 | 91.7 ± 0.5 |
| | Pushpins | **90.3** | 58.1 | 77.6 | 74.9 | 78.9 ± 3.7 |
| | Screw bag | 87.0 | 59.8 | 86.6 | 70.5 | **92.4 ± 1.1** |
| | Splicing connectors | **96.8** | 57.1 | 68.7 | 78.3 | 78.3 ± 0.3 |
| | Avg. Structural | **88.3** | 66.9 | 78.9 | 80.7 | 85.2 ± 0.7 |
| | Avg. Total | 77.4 | 68.9 | 74.0 | 83.4 | **88.3 ± 0.7** |

## B   Set descriptor comparison

**Clustering-based set descriptors.** We compare our histogram-based approach to the VLAD and Bag-of-Features approaches. We report our results in Tab.8. We use the number of means $K = 100$ cluster, but this result persists when we varied the number of clusters. We do not report the results on Fisher-Vectors as the underlying GMM model (unlike K-means) requires unfeasible computational resources with our set dimensions. Taken together, it seems that the underlying clustering assumption does not fit the sets we wish to describe as well our set descriptors.

$k$**NN versus distance to the mean.** We found that using the Gaussian model only to whiten the data and taking the distance to the 1 nearest neighbors (and not to the center) worked better for the MVTec-LOCO dataset (see Tab.8). The nearest neighbors density estimation algorithm better models the density distribution when the Gaussian assumption is not an accurate description of the data.

Table 7: UEA datasets, average ROC-AUC (%) over all classes including error bounds

|      | OCSVM | IF | LOF | RNN | LSTM-ED |
|------|-------|-----|-----|-----|---------|
| EPSY | 61.1 | 67.7 | 56.1 | $80.4 \pm 1.8$ | $82.6 \pm 1.7$ |
| NAT | 86 | 85.4 | 89.2 | $89.5 \pm 0.4$ | $91.5 \pm 0.3$ |
| SAD | 95.3 | 88.2 | 98.3 | $81.5 \pm 0.4$ | $93.1 \pm 0.5$ |
| CT | 97.4 | 94.3 | 97.8 | $96.3 \pm 0.2$ | $79.0 \pm 1.1$ |
| RS | 70 | 69.3 | 57.4 | $84.7 \pm 0.7$ | $65.4 \pm 2.1$ |
| Avg. | 82.0 | 81.0 | 79.8 | 86.5 | 82.3 |

|      | DeepSVDD | DAGMM | GOAD | DROCC | NeuTraL | Ours |
|------|----------|-------|------|-------|---------|------|
| EPSY | $57.6 \pm 0.7$ | $72.2 \pm 1.6$ | $76.7 \pm 0.4$ | $85.8 \pm 2.1$ | $92.6 \pm 1.7$ | $\mathbf{98.1} \pm 0.3$ |
| NAT | $88.6 \pm 0.8$ | $78.9 \pm 3.2$ | $87.1 \pm 1.1$ | $87.2 \pm 1.4$ | $94.5 \pm 0.8$ | $\mathbf{96.1} \pm 0.1$ |
| SAD | $86.0 \pm 0.1$ | $80.9 \pm 1.2$ | $94.7 \pm 0.1$ | $85.8 \pm 0.8$ | $\mathbf{98.9} \pm 0.1$ | $97.8 \pm 0.1$ |
| CT | $95.7 \pm 0.5$ | $89.8 \pm 0.7$ | $97.7 \pm 0.1$ | $95.3 \pm 0.3$ | $99.3 \pm 0.1$ | $\mathbf{99.7} \pm 0.1$ |
| RS | $77.4 \pm 0.7$ | $51.0 \pm 4.2$ | $79.9 \pm 0.6$ | $80.0 \pm 1.0$ | $86.5 \pm 0.6$ | $\mathbf{92.3} \pm 0.3$ |
| Avg. | 81.1 | 74.6 | 87.2 | 86.8 | 94.4 | $\mathbf{96.8}$ |

Table 8: MVTec-LOCO ablation: using no raw-pixels level. ROC-AUC (%).

|          | Mahalanobis (dist. to $\mu$) | BoF | VLAD | Ours |
|----------|------------------------------|------|------|------|
| Breakfa. | 93.6 | 84.7 | 87.9 | $\mathbf{97.6}$ |
| Juice bo. | 91.6 | 93.8 | $\mathbf{97.5}$ | 97.0 |
| Pushpins | 79.9 | 78.2 | 79.1 | $\mathbf{88.6}$ |
| Screw b. | 68.2 | 69.9 | 64.1 | $\mathbf{81.7}$ |
| Splicing. | 78.2 | 85.0 | 89.7 | $\mathbf{91.1}$ |
| Average | 82.3 | 82.3 | 83.7 | $\mathbf{91.2}$ |

# C    Implementation Details

*Histograms.* We use the cumulative histograms as our set features for both data modalities (of Sec.3.3).

## C.1    Image Anomaly Detection

*Parameters.* For the image experiments, we use histograms of $K = 5$ bins and $r = 1000$ projections. For the raw-pixels layer, we used a projection dimension of $r = 10$ and no whitening due to the low number of channels. To avoid high variance between runs, we did 32 different repetitions for the raw-pixel scoring and used the median score. We use $k = 1$ for the $k$NN density estimation.

*Preprocessing.* Before feeding each image sample to the pre-trained network we resize it to $224 \times 224$ and normalize it according to the standard ImageNet mean and variance.

Considering that classes in this dataset are provided in different aspect ratios, and that similar objects may look different when resized to a square, we found it beneficial to pad each image with empty pixels. The padded images have a $1 : 1$ aspect ratio, and resizing them would not change the aspect ratio of the featured objects.

*Software.* For the whitening of image features we use the *ShrunkCovariance* function from the *scikit-learn library* Pedregosa et al. (2011) with its default parameters. For $k$NN density estimation we use the *faiss* library Johnson et al. (2019).

*Computational resources.* The experiments were run on a single RTX2080-GT GPU.

Table 9: Robustness to the choice of $\lambda$. Average ROC-AUC (%) on logical anomalies classes.

| $\lambda$ | 0.2 | 0.1 (Ours) | 0.05 | 0.02 |
|---|---|---|---|---|
| | 90.2 | 91.2 | 91.4 | 90.7 |

Table 10: Ablation for values of $r$, the number of random projections. Average ROC-AUC on MVTec-LOCO, logical anomalies. $K = 5$, $\sigma = 0.6$ (%).

| $r$ | 2000 | 1000 | 500 | 200 | 100 |
|---|---|---|---|---|---|
| Avg. Logical | 91.2 | 91.2 | 90.6 | 89.6 | 86.1 |

### C.2 Time Series Anomaly Detection

*UEA Experiments.* We used each time series as an individual training sample. We chose a kernel size of $\tau = 9$, $r = 100$ projection, $K = 20$ quantiles, and a maximal number of pyramid levels of $L = 10$, each using consecutive strides. The results varied only slightly within a reasonable range of the hyperparameters. E.g. using 5, 10, 15 levels yielded an average ROCAUC of 97, 96.8, 96.8 across the five UEA datasets.

*Padding.* Prior to window extraction, the series $x$ is first right and left zero-padded by $\frac{\tau}{2}$ to form a padded series $x'$. The first window $w_1$ is defined as the first $\tau$ observations in padded series $S'$, i.e. $w_1 = x'_1, x'_2..x'_\tau$. We further define windows at higher scales $W^s$, which include observations sampled with stride $c$. At scale $c$, the original series $x$ is right and left zero-padded by $\frac{c \cdot \tau}{2}$ to form padded series $S'^c$.

*Spoken Arabic Digits processing* We follow the processing of the dataset as done by Qiu et al. Qiu et al. (2021). In private communications the authors explained that only use sequences of lengths between 20 and 50 time steps were selected. The other time series were dropped.

*Computational resources.* The experiments were run on a modest number of CPUs on a computing cluster. The baseline methods were run on a single RTX2080-GT GPU.

## D  Logical Anomaly Detection Robustness

We check the robustness of our results to the parameter $\lambda$ - the weighting between the raw-pixels level anomaly score to the anomaly score derived from the ResNet features (Sec.5.3). As can be seen in Tab.9, our results are robust to the choice of $\lambda$.

## E  Further Image Anomaly Detection Ablation

We include here the ablation tables for the number of random projections and number of bins, for logical anomaly detection (Tab.10,11).

## F  Time Series Anomaly Detection Ablations

**Number of projections.** Using a high output dimension for projection matrix $P$ increases the expressively but also increases the computation cost. We investigate the effect of the number of projections on the final accuracy of our method. The results are provided in Fig. 4. We can observe that although a small number of

Table 11: Ablation for values of $K$, the number of bins per histogram. Average ROC-AUC on MVTec-LOCO, logical anomalies. $r = 1000$, $\sigma = 0.6$ (%).

| K | 20 | 10 | 5 | 4 | 3 | 2 |
|---|---|---|---|---|---|---|
| Avg. Logical | 91.1 | 91.3 | 91.2 | 91.2 | 90.8 | 90.2 |

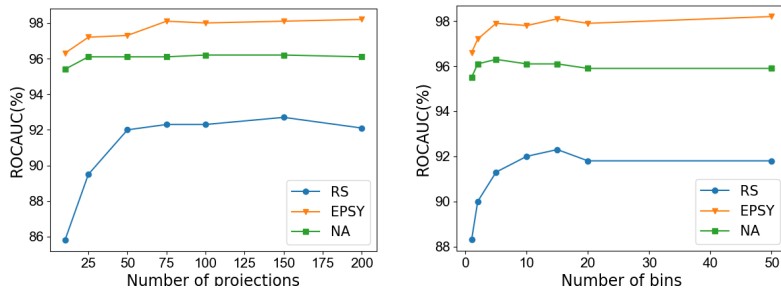

Figure 4: Ablation of accuracy vs. the number of projections (left) and the number of bins (right) for different time series datasets.

Table 12: An ablation of distance calculation methods for different time series datasets. ROC-AUC (%).

|  | EPSY | RS | NA | CT | SAD |
|---|---|---|---|---|---|
| Quantized SWD (No whitening) | 62.1 | 70.9 | 93.6 | 98.5 | 78.8 |
| SWD | 90.7 | 84.8 | 91.9 | 99.3 | 88.4 |
| With Whitening | **98.1** | **92.3** | **96.1** | **99.7** | **97.8** |

projections hurts performance, even a moderate number of projections is sufficient. We found 100 projections to be a good tradeoff between performance and runtime.

**Number of bins.** We compute the accuracy of our method as a function of the number of bins per projection. Our results (Fig. 4) show that beyond a very small number of bins, a larger number of bins does not help. We found 20 bins to be sufficient in all our experiments.

**Effect of Gaussian density estimation.** We ablate the use of Gaussian modeling in Tab. 12 (Quantized SWD). We can see that our approach achieves far better results, attesting to the importance of modeling the correlation between projections.

**Comparison to the Sliced Wasserstein distance.** In Sec.3.5 we highlight the connection between our approach and the Sliced Wasserstein distance. An empirical comparison between the approaches can be found in Tab. 12. Our results show that computing the SWD without histogram binning can be much more accurate than with binning (Quantized SWD). However, the binning is necessary for our whitening technique (Sec.3.4), which significantly outperforms standard SWD. We also note that increasing the number of bins (making the quantization finer) does not improve the accuracy of our full approach.

**Comparing projection sampling methods.** We compare three different projection selection procedures: (i) Gaussian: sampling the weights in $P$ from a random Normal Gaussian distribution (ii) Using an identity projection matrix: $P = I$ . (iii) PCA: selecting $P$ from the eigenvectors of the matrix containing all (raw) features of all training windows. PCA selects the projections with maximum variation but is computationally expensive. The results are presented in Tab. 13. We find that the identity projection matrix under-performed the other approaches (as it provides no variable mixing). Surprisingly, we do not see a large difference between PCA and random projections.

Table 13: An ablation of projection sampling methods for different time series datasets. ROC-AUC (%).

|  | EPSY | RS | NA | CT | SAD |
|---|---|---|---|---|---|
| Id. | 97.1 | 90.2 | 91.8 | 98.2 | 78.3 |
| PCA | **98.2** | 91.6 | 95.8 | **99.7** | 96.7 |
| Rand | 98.1 | **92.3** | **96.1** | **99.7** | **97.8** |

Table 14: An ablation of time-series number of pyramid levels. ROC-AUC (%), $L = 9$.

| $\tau$ | 5 | 8 | 10 (Ours) | 12 | 15 |
|---|---|---|---|---|---|
| Avg. time-series | 96.7 | 96.9 | 96.8 | 96.8 | 96.7 |

Table 15: An ablation of time-series window size. ROC-AUC (%), $\tau = 10$.

| $L$ | 5 | 7 | 9 (Ours) | 11 | 13 |
|---|---|---|---|---|---|
| Avg. time-series | 96.8 | 96.8 | 96.8 | 96.8 | 96.6 |

**Effect of number of pyramid levels and window size.** We ablate the two hyperparameters of the time-series feature extraction: the number of pyramid windows used $L$, and the number of samples per window $\tau$ (see Sec.4.2). We find that in both cases the results are not sensitive to the chosen parameters (Tab.14,15).

## G   Using the Central Limit Theorem for Set Anomaly Detection

We model the features of each window $f$ as a normal set as IID observations coming from a probability distribution function $p(f)$. The distribution function is *not* assumed to be Gaussian. Using a Gaussian density estimator trained on the features of elements observed during training is unlikely to be effective for element-level anomaly detection (due to the non-Gaussian $p(f)$).

An alternative formulation to the one presented in section 3, is that each feature $f$ is multiplied by projection matrix $P$, and then each dimension is discretized and mapped to a one-hot vector according to its relevant histogram bin. This formulation therefore maps the representation of each element to a sparse binary vector. The mean of this one-hot vector representation of elements in the set recovers the normalized histogram descriptor precisely (therefore this formulation is equivalent to the one in section 3). As the histogram is a mean of the one-hot representations of elements, it has superior statistical properties. In particular, the Central Limit Theorem states that under common conditions the sample mean follows the Gaussian distribution. While typically in anomaly detection only a single sample is presented at a time, the situation is different when treating samples as sets. Although the elements are often not IID, given a multitude of elements, an IID approximation may still be applicable. This explains the high effectiveness of Gaussian density estimation in our formulation.

## H   Further Discussion

**Incorporating deep features for time series data.** Our method outperforms the state-of-the-art in time series anomaly detection without using deep neural networks. While this is an interesting and surprising result, we believe that deep features will be incorporated into similar approaches in the future. One direction for doing this is replacing the window projection features with a suitable deep representation, while keeping the set descriptors and Gaussian modeling steps unchanged.

**Fine-tuning deep features for anomaly detection.** Following recent works in anomaly detection and anomaly segmentation, we used fixed pre-trained features as the backbone of our method. Although some methods fine-tune deep features for anomaly detection based on the normal-only training set, we keep them constant. Doing so allows an interpretable examination of the relative strength of our novel scoring function with respect to prior works that use fixed features. Yet, we expect that fine-tuning such features could lead to further gains in the future.

