# OpenReview forum: "Set Features for Anomaly Detection"
_TMLR — Accepted by TMLR_

### Review · Reviewer_UZyq · 2024-07-23

**Summary Of Contributions:**

This paper addresses anomaly detection where anomalies are characterized by unusual combinations of normal elements. The authors propose set features that model each sample based on the distribution of its elements. Experiments across several datasets demonstrate the effectiveness of the proposed model.

**Audience:**

Yes

**Broader Impact Concerns:**

None.

**Claims And Evidence:**

Yes

**Requested Changes:**

Please see the weaknesses.

**Strengths And Weaknesses:**

Strengths:
1. Anomaly detection is a critical task warranting thorough investigation.
2. Modeling sample distributions shows promise and effectiveness in anomaly detection.


Weaknesses:
1. The Introduction should clarify existing solutions to this issue and compare them with the proposed approach.
2. In the experiments, it is important for all baselines to report standard deviations to assess their robustness. Additionally, comparing with recent baselines used in time series anomaly detection would strengthen the evaluation.

---

> ### Author Response · Authors · 2024-08-27
> **Authors Response**
>
> We thank Reviewer UZyq for the constructive comments and feedback. We would like to address the concerns below:
>
> ***"The Introduction should clarify existing solutions to this issue and compare them with the proposed approach."***
>
> We assume the reviewer refers to the issue of “logical anomalies”. We added a short paragraph to the introduction discussing alternative and complementary solutions to ours.
>
> ***"it is important for all baselines to report standard deviations to assess their robustness"***
>
> We enthusiastically agree! For this reason we reported standard deviations to our method, as well as to the reproduced top-performing  EfficientAD, and the ensembles of our two methods (Tab.6). We also present standard deviation for many methods in App.Table 8. We believe that all papers should publish standard deviation estimates; yet, it is impractical for us to estimate the standard deviations for all prior methods, especially as many of them have not released their code.
>
> ***"...comparing with recent baselines used in time series anomaly detection would strengthen the evaluation"***
> We have compared to 10 different baselines which we found as the most relevant to the task we evaluate. If the reviewer thinks any specific baseline is missing, we are happy to include it. We emphasize that most recent baselines deal with *point-in-time anomaly detection*: finding specific times within a series that contain an anomaly, rather than identifying the entire series as normal or anomalous. (See Sec.1; revised Sec.7).
>
>
> Thank you once again for your comments! We are happy to continue the discussion at any time until the end of the discussion period.

---

### Review · Reviewer_HGet · 2024-07-29

**Summary Of Contributions:**

**Summary**: This paper proposes a novel method for anomaly detection using set features, where anomalies are identified based on unusual combinations of normal elements. Specifically, the authors introduce a technique that models each sample as a distribution of its elements and uses random projection histograms for feature representation, followed by density estimation for anomaly scoring. The proposed method is evaluated on two benchmark datasets against several baseline algorithms.

**Audience:**

Yes

**Broader Impact Concerns:**

None.

**Claims And Evidence:**

Yes

**Requested Changes:**

As in Weaknesses, in order to meet the criteria of a top-tier venue such as TMLR, the authors need to

1. make major revisions to section 3 to improve the presentation of the methodology
2. revise section 5 to make a more comprehensive evaluation

**Strengths And Weaknesses:**

**Strengths**:

1. The set representation for anomaly detection is novel to me.
2. Evaluating the method on the time series task is important.


**Weaknesses**:

1. It seems that the proposed method is only applicable to a very specific case of anomaly detection, where the combinations of elements are unusual. The authors should at least discuss the applicability of the proposed method in other scenarios.
2. There are too many notations that are not properly defined. For example, when it comes to a set, we usually use curly brackets. In Section 3.2, it first use $x1, x2… x_{N_S} $ and then $[e_1, e_2..e_{N_E}]$ to define sets. In addition, $ e_1, e_2..e_{N_E}$ are not formally defined at all. Therefore, it is hard for me to understand the formulation from the very beginning. Due to a similar reason, it is also hard for me to understand the histogram descriptor in Section 3.3 (e.g., $s[j]$, $f[j]$).
3. The Mahalanobis distance is adopted when computing the anomaly score in Section 3.4. The implicit assumption here is the histogram projection follows a Gaussian distribution, which does not necessarily hold in practice. The authors should at least have some discussions on it.
4. The proposed method is only evaluated on two datasets. Other benchmarks (e.g.,VisA), should also be considered.

Overall, I don't think this work is ready to be published in a top journal such as TMLR.

---

> ### Author Response · Authors · 2024-08-27
> **Authors Response**
>
> Thank you for your detailed review. We appreciate that you found our method novel and the task we study important. Please find our response below:
>
> ***"...only applicable to a very specific case of anomaly detection, where the combinations of elements are unusual"***
>
>
> Our method indeed focuses on the specific, yet important, case of logical anomaly detection. There is a growing interest within the community in this type of anomalies. While our method does not cover structural image anomalies, it can be easily combined with other methods (as demonstrated in Table 2).
>
> Although structural anomalies represent a larger research community, we think that introducing a novel anomaly detection algorithm, applicable to both image-level logical anomalies and time-series sequence-level anomalies, is a significant contribution. We thank the reviewer for highlighting the need for clarification here and have added to section 7 a discussion regarding the cases where the method is or isn’t applicable.
>
>
> ***"use curly brackets”, “e1,e2..eNE are not formally defined … (e.g., s[j], f[j])"***
> We thank the reviewer for this suggestion and updated the manuscript accordingly.
>
>
> ***"The implicit assumption here is the histogram projection follows a Gaussian distribution"***
> We would like to refer the reviewer to Appendix H where we discuss the Gaussian assumption. We added a clearer reference to it from the main text.
>
> ***"The proposed method is only evaluated on two datasets. Other benchmarks (e.g.,VisA), should also be considered."***
>
>
> We thank the reviewer for their suggestion but respectfully disagree with this point. Our method is evaluated on (i) MVTec - LOCO, a benchmark composed of 5 different sub-tasks, where each sub-task contains numerous different types of anomalies. (ii) 5 different time series datasets.
>
> Other benchmarks only evaluate structural anomaly detection, which are not the focus of our method. We hope our revision of section 7 will better clarify this point.
>
> Thank you again for your comments! We are happy to continue the discussion at any time until the end of the discussion period. Thank you!

---

### Review · Reviewer_p7s6 · 2024-08-23

**Summary Of Contributions:**

This work proposes a variant of a random projection method for anomaly detection; the method applies these projections to features extracted from pretrained deep neural networks in images and simple window pyramids in time series, and (effectively) uses a model of the projections as Gaussian for density estimation. It obtains strong performance on a benchmark for "logical" anomalies that had largely eluded previous efforts. The paper also exhibits a number of ablations evaluating the contribution of various components and hyperparameter settings to these results.

**Audience:**

Yes

**Broader Impact Concerns:**

There is no broader impact statement, but I am not particularly concerned about this for this work.

**Claims And Evidence:**

Yes

**Requested Changes:**

Following the above, my main request is a precise and complete description of the method, preferably using complete pseudocode or mathematical definitions of the components, in particular the histograms.

**Strengths And Weaknesses:**

The main strength of the paper is the quality of the empirical results obtained, in particular on the logical anomalies in the MVTec-LOCO benchmark. The ablations are also helpful. The discussion in Appendix H helps motivate why it is particularly reasonable to use a Gaussian density estimation model in the context of random-projection-based methods.

The main weakness is that the method's description is vague and confusing. In particular, the paper claims that there are two main distinctions from the closely related prior work by Pevny, LODA, using a different density estimator and "using sets of multiple elements rather than single sample descriptions". While there is some further, helpful discussion of the difference in the density estimator in Appendix G/Table 13, I don't fully understand this "sets of elements" description; the paragraph "Histogram descriptor" in Section 3.3 is especially confusing. Section 3.2 says each example is described by a set of elements (patches or windows), but then in 3.3 there's a set of values for each dimension. What do the dimensions range over -- does this correspond to the elements of section 3? But when you refer to it as a set, it sounds like you mean that it is unordered -- if so, how does this indexing make sense? Are the projections simply computed on some raw representation of the data (interpreting patches as flattened into vectors of pixels, for example)?

---

> ### Author Response · Authors · 2024-08-27
> **Authors Response**
>
> We sincerely thank the reviewer for their constructive comments and valuable feedback on our submission. We think the clarity issues the reviewer raised were fair. We revised them in the manuscript (Sec 3.2, 3.3) and explained them point-by-point below.
>
> ***"Section 3.2 says each example is described by a set of elements (patches or windows), but then in 3.3 there's a set of values for each dimension. What do the dimensions range over -- does this correspond to the elements of section 3?"***
>
> We indeed examine each sample as a set of $N_E$ elements (patches or temporal window). We describe each element using an $N_D$ dimensional feature.  The set s[j] is the set of the values of the j`th (j between 1 and $N_D$) feature dimension of each element in the sample, i.e., a set of $N_E$ scalars.
>
> We updated our manuscript with a clearer description and an algorithm box, and would be happy to add further clarifications if needed.
>
> ***"What do the dimensions range over -- does this correspond to the elements of section 3? But when you refer to it as a set, it sounds like you mean that it is unordered -- if so, how does this indexing make sense?"***
>
> Indeed, indexing is confusing for sets (although in practice, we implemented the set as an array, and it made sense there). Instead, we revised the manuscript to remove all indexing of elements. Now the index refers to the feature dimension only.
>
>
> ***"Are the projections simply computed on some raw representation of the data (interpreting patches as flattened into vectors of pixels, for example)?"***
>
> We embed each element to a feature representation using a deep network for images (Sec.4.1) of classical features for time series (Sec.4.2). We added a clarification to the manuscript.
>
>
> Thank you again for your excellent comments. We are happy to continue the discussion at any time until the end of the discussion period. Thank you!

---

### Review · Reviewer_aQQj · 2024-10-18

**Summary Of Contributions:**

The paper aims to develop a novel method for detecting anomalies in images or time series data, specifically targeting what the authors term "_logical_ anomalies." These anomalies arise from unusual combinations of items rather than from the appearance of an unusual item itself. The proposed method simplifies bags of local features for computational efficiency by projecting them into 1D space and then quantizing them to obtain histograms.

**Audience:**

Yes

**Broader Impact Concerns:**

-

**Claims And Evidence:**

Yes

**Requested Changes:**

## Critical Adjustments

1. To strengthen the paper, a quantitative comparison between the proposed method and SW distance without quantization should be conducted. This would help validate the claim that histograms indeed offer an advantage.
2. The authors should clarify their statement about quantization in Section 3.5 to avoid any misleading information for readers.

## Non-Critical Adjustment

Correct all typos in the text, such as "article amsmath algorithm algorithmic" on page 6.

**Strengths And Weaknesses:**

## Strengths

- The paper introduces a novel approach to anomaly detection that focuses on logical anomalies, which is an important aspect often overlooked in existing methods.
- The use of projections and quantization simplifies the feature representation, making the method computationally efficient.

## Weaknesses

- There is uncertainty about whether simplifying the feature representations by using histograms actually improves performance, given that Sliced Wasserstein (SW) distance can already efficiently operate on discrete distributions (sums of Diracs) without requiring quantization into histograms.
- The presentation in Section 3.5 seems misleading as it suggests the necessity for quantization for SW when, in reality, SW could be directly applied to raw distributions of features.

---

> ### Author Response · Authors · 2024-10-28
> **Authors Response**
>
> We thank the reviewer for their valuable feedback. We appreciate that the reviewer found our method novel and the task we study important. Please find our response below:
>
> ***“simplifying the feature representations by using histograms actually improves performance, given that Sliced Wasserstein (SW) distance can already efficiently operate on discrete distributions”***
>
> ***“To strengthen the paper, a quantitative comparison between the proposed method and SW distance without quantization should be conducted.”***
>
> We agree that one can use the sliced Wasserstein distance (SWD) directly on the projected values without quantization. While the Sliced Wasserstein defines a distance function between pairs on sets, it is not natively a feature representation. The feature embedding of sets allows us to use a whitening matrix and calculate the Mahalanobis distance. This is an important step as Table 13 demonstrates. However, we agree that calculating the Sliced Wasserstein distance directly (without quantization) indeed has some advantages compared to our histogram methods. Yet, this is only in the case where we do not use the feature whitening. We updated Table 13 and App.G in the revised manuscript accordingly. Please also find the revised table below:
>
> |                             | EPSY |  RS  |  NA  |  CT  | SAD  |
> |-----------------------------|------|------|------|------|------|
> | Quantized SWD (No whitening) | 62.1 | 70.9 | 93.6 | 98.5 | 78.8 |
> | SWD                         | 90.7 | 84.8 | 91.9 | 99.3 | 88.4 |
> | Quantized With Whitening          | **98.1** | **92.3** | **96.1** | **99.7** | **97.8** |
>
>
> ***“The authors should clarify their statement about quantization in Section 3.5 to avoid any misleading information for readers”***
>
> We added in Sec.3.5 a paragraph explaining that the quantization is needed for our approach but not for the sliced Wasserstein distance.
>
>
> Thank you again for your suggestions and comments!

---

> > ### Comment · Reviewer_aQQj · 2024-10-29
> > **Thanks for your answer**
> >
> > I would like to thank the authors for their answer that fully covers my comments. I hence change my recommendation to lean towards acceptance.

---

> > > ### Author Response · Authors · 2024-12-05
> > >
> > > Thank you for acknowledging our rebuttal and for taking the time to review our work!
> > > Your insights have been instrumental in shaping the final version of our submission.

---

### Author Response · Authors · 2024-08-27
**Global Response**

We thank all the reviewers for their valuable feedback. Our work introduces SINADB, an anomaly detection algorithm focused on identifying logical time-series and image anomalies, pushing the state-of-the-art in these areas.

We appreciate that the reviewers acknowledge our promising results (*p7s6*, *UZyq*),the importance of the studied task (*HGet*, *UZyq*), and our approach as novel (*HGet*) and well-motivated (*p7s6*).

Following your suggestions, we include a revised manuscript with a clearer explanation of our method, its limitations, and its relation to existing methods. We also added an algorithm box better describing our method. Text changes with respect to the original submission are highlighted in green.

Please find responses to specific concerns below.

---

### Decision · Action_Editor_byoE · 2024-11-04

**Recommendation:** Accept as is

**Comment:**

The authors propose an approach to improve anomaly detection when the anomaly is the result of anomalous combinations of features, and not of single anomalous features. Experiments that include ablation studies validate the claims. The proposed method could be of interest for those studying anomalies of sets of objects and other logical anomalies.

**Audience:**

The proposed method could be of interest for those studying anomalies of sets of objects and other logical anomalies.

**Claims And Evidence:**

The authors propose an approach to improve anomaly detection when the anomaly is the result of anomalous combinations of features, and not of single anomalous features. Experiments that include ablation studies validate the claims.

---

> ### Author Response · Authors · 2024-12-05
>
> We sincerely thank the area chair and all reviewers for their time and effort invested in reviewing our paper.
>
> Below is a summary of the main updates we made to the camera ready:
>
> - We performed an additional spell check and revised the appendix to avoid repetitive content.
> - We added a table of contents to the appendix.
> - We included an acknowledgment section.
> - We deanonymized the paper and provided a link to our official code.
>
> Once again, we sincerely thank the area chair and all the reviewers!